# AgentStore: Scalable Integration of Heterogeneous Agents As Specialized Generalist Computer Assistant

## Abstract

Digital agents capable of automating complex computer tasks have attracted considerable attention due to their immense potential to enhance human-computer interaction. However, existing agent methods reveal deficiencies in their generalization and specialization capabilities, especially in handling open-ended computer tasks in real-world environments. Inspired by the rich functionality of the App store, we present **AgentStore**, a scalable platform designed to dynamically integrate heterogeneous agents for automating computer tasks. AgentStore empowers users to integrate third-party agents, allowing the system to continuously enrich its capabilities and adapt to rapidly evolving operating systems. Additionally, we propose a novel core **MetaAgent** with the **AgentToken** strategy to efficiently manage diverse agents and utilize their specialized and generalist abilities for both domain-specific and system-wide tasks. Extensive experiments on challenging benchmarks demonstrate that AgentStore surpasses the limitations of previous systems with narrow capabilities, particularly achieving a significant improvement from 11.21% to 23.85% on the OSWorld benchmark, more than doubling the previous results. Comprehensive quantitative and qualitative results further demonstrate AgentStore's ability to enhance agent systems in both generalization and specialization, underscoring its potential for developing the specialized generalist [1] computer assistant. All our codes will be made publicly available.

## 1 Introduction

The continual evolution of computer Operating Systems (OS), along with proliferating applications, has transformed how people work and live. This transformation goes beyond daily life like shopping and gaming, encompassing professional works such as writing in Office or editing in Photoshop. However, this increased functionality comes with a steep learning curve, often burdening users. As a result, autonomous computer assistants—once limited to fiction like *JARVIS in Iron Man or MOSS in Wandering Earth*—have become a concrete pursuit, attracting great interest from researchers.

Advancements in Multimodal Large Language Models (MLLMs) (OpenAI, 2023; Reid et al., 2024), are gradually turning this vision into reality. MLLM-based agents have already demonstrated remarkable intelligence in handling complex tasks, benefiting from their strong capabilities in planning and reasoning (Wei et al., 2022; Yao et al., 2023). Following this trend, using MLLMs to build digital agents for automating computer tasks has become a promising direction (Zhang et al., 2024a). However, real-world OS environments encompass a diverse array of open-ended computer tasks, each with inherent requirements for capabilities across multi-dimensions (Xie et al., 2024), posing substantial challenges to existing methods. Specifically, "Task_1" in Figure 1 illustrates that many computer tasks necessitate specific knowledge and operations. In such scenarios, existing generalist agents (Wu et al., 2024; Tan et al., 2024) often underperform due to their lack of these specialized abilities. Conversely, specialized agents, despite excelling at specific tasks within single domains like tabular data processing (Li et al., 2024; Chen et al., 2024a) or web browsing (Zhou et al., 2023; Deng et al., 2024), cannot generalize across different applications or broader system en-

---

[1]The concept of the "Specialized Generalist" refers to an AI system that excels in specific tasks, surpassing human experts, while still maintaining broad general capabilities (Zhang et al., 2024b).

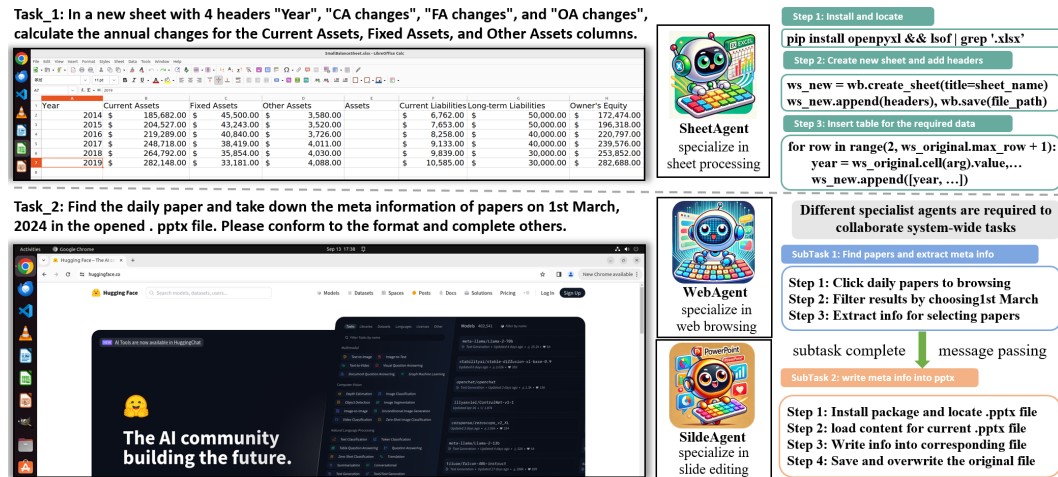

Figure 1: Task examples illustrate that diverse open-ended tasks require a combination of generalization and specialization capabilities. The right part provides a simple overview of specific steps.

vironments. Therefore, these agents struggle to perform independently when confronted with more integrated, system-wide tasks like "Task_2" in Figure 1. This heterogeneous demand for capabilities across various tasks presents a challenge for existing single generalist or specialized agents.

We attribute this dilemma to overlooking a key factor behind the success of modern operating systems: App store[2]. As a distribution platform, the App store provides an ever-expanding set of functionalities that extend beyond the core OS itself. Correspondingly, we argue that *specialized generalist computer agents should possess the characteristics akin to those of the App store, evolving to grow heterogeneous abilities and autonomously handle an increasingly diverse range of tasks*. To substantiate this, we propose **AgentStore**, a flexible and scalable platform for dynamically integrating various heterogeneous agents to independently or collaboratively automate OS tasks (illustrated on the right in Figure 1). AgentStore allows users to quickly integrate their own specialized agents into the platform, similar to the functionality of the App store. This scalable integration allows the framework to dynamically adapt itself to the evolving OS, providing the multi-dimensional capabilities needed for open-ended tasks, and ultimately offering a robust solution for developing the specialized generalist computer assistant.

Specifically, we first develop a prototype of AgentStore, establishing an agent integration protocol and creating over 20 agents with diverse functionalities to handle a wide range of OS tasks across widely used desktop applications. Based on this foundation, the main challenge is efficiently managing the rapidly growing and increasingly large number of agents, which overwhelms traditional management methods, such as In-Context Learning (ICL; Dong et al., 2022) and full Fine-Tuning (FT; Qin et al., 2023). To this issue, we introduce a novel MLLM-based **MetaAgent** with **Agent-Token** strategy, to select the most suitable agent(s) to independently or collaboratively complete tasks. Specifically, each integrated agent in AgentStore is denoted as a learnable token embedding in MetaAgent's architecture like a word token embedding. During inference, MetaAgent activates the corresponding agent to execute the task when an agent token is predicted. Innovatively, we enhance this approach by shifting from single-token (Hao et al., 2024) to multi-token prediction, allowing MetaAgent to predict and coordinate multiple agents for collaborative task execution. Additionally, we propose an automated process with self-instruct for tuning AgentToken without relying on manual data, further enhancing AgentStore's practicality in real-world scenarios.

We validate the effectiveness of AgentStore through extensive experiments in OS environments. On the highly challenging OSWorld benchmark, a real-world computer environment with 369 tasks, AgentStore achieved a success rate of 23.85%, more than doubling the performance of the previous best system (11.21%) (Xie et al., 2024). Comprehensive quantitative and qualitative results, along with ablation studies, highlight the critical importance of scalable heterogeneous agent integration in expanding the system's capabilities. Similar outcomes were observed when evaluating AgentStore in a mobile environment, demonstrating our approach's adaptability for automating tasks across multiple OS platforms. Additionally, we demonstrated the broad applicability of the AgentToken

---

[2]In this paper, App store not only refers to the App Store for Apple but all similar platforms. See the specific concept in App store.

paradigm in comparison to other strategies, highlighting its efficiency in training and its effectiveness in dynamically managing agents within AgentStore. We conclude our contributions as follows:

- **AgentStore**: We propose a scalable platform for dynamically integrating heterogeneous agents to automate operating system tasks. AgentStore adapts itself to evolving environments, offering a robust solution for developing specialized generalist computer assistants.

- **MetaAgent with AgentToken**: We introduce MetaAgent to manage the growing number of agents and propose AgentToken to enhance training efficiency and enable plug-and-play functionalities.

- **Stunning Results**: AgentStore achieves SOTA results on challenging benchmarks, more than doubling the performance of previous systems. Our comprehensive analysis demonstrates how AgentStore expands agent capabilities in both generalization and specialization.

## 2 RELATED WORK

**LLM-based Agents.** Recent advancements in (M)LLMs (OpenAI, 2023; Reid et al., 2024) have led to the development of highly capable AI agents, applied across various domains, including robotics (Driess et al., 2023), software development (Wang et al., 2024), and beyond. A rapidly growing research field among these is automating interactions with computer environments to solve complex tasks. Early work primarily focused on specific scenarios, such as web manipulation (Yao et al., 2022; Deng et al., 2024), command-line coding (Sun et al., 2024), and gaming (Wang et al., 2023a). Following this, more recent methods (Wu et al., 2024; Tan et al., 2024) have started exploring general-purpose computer agents capable of interacting with diverse components of an operating system. Unfortunately, both of these struggle with open-ended tasks in real environments, exposing limitations in their generalization and specialization capabilities. To address these shortcomings, this paper introduces AgentStore to build the specialized generalist computer assistant.

**Multi-Agent Systems.** Recently, various approaches (Park et al., 2023; Sun et al., 2023; Wu et al., 2023; Hong et al., 2023) have been proposed to facilitate effective collaboration and communication among multi-agent to overcome hallucinations, ensuring deterministic and trustworthy results.

While these approaches have shown promising results in domains such as automating coding, they still exhibit two major limitations. First, by using a fixed number of agents with predefined roles, *they lack support for dynamically integrating agents*. Second, *their agents are usually homogeneous*, which limits agent diversity and consequently constrains their range of capabilities. Therefore, our approach is designed to support the dynamic integration of a large number of third-party agents to leverage their advantages in quantity and diversity. AgentStore expands the capability boundaries of current multi-agent systems.

## 3 AGENTSTORE

We first provide a comprehensive overview and detail key components of the framework in Section 3.1. Then, Section 3.2 introduces MetaAgent, explaining how to effectively manage the rapidly growing and large number of agents via AgentToken. Finally, Section 3.3 details how AgentToken can be efficiently trained using an automated process with self-instruct.

### 3.1 FRAMEWORK OVERVIEW

As illustrated in Figure 2, AgentStore consists of three main components: AgentPool, AgentEnroll, and MetaAgent. The AgentPool stores all feature-specific agents with distinct functionalities. AgentEnroll defines the integration protocol for adding new agents to the AgentPool. Finally, the MetaAgent selects the most suitable agent(s) from AgentPool to independently or collaboratively complete tasks. In this section, we provide a detailed explanation of these key components.

**AgentPool:** The AgentPool is a collection of all available agents within AgentStore. To build the prototype of AgentStore, we organized over 20 agents within AgentPool, each with distinct functionalities. These agents range from unimodal to multimodal, from open-source to closed-source models, and from Command-Line Interfaces (CLI) to Graphical User Interfaces (GUI). The

Figure 2: The illustration on the main components in AgentStore.

diverse capabilities of these agents cover common applications and tasks in both daily life and professional work. This heterogeneous combination provides a solid foundation to validate the effectiveness of the AgentStore concept. The details of these agents are presented in Appendix A.

**AgentEnroll:** When a developer creates a new OS agent and seeks to integrate it into AgentStore, it is essential to register the agent's information in a standardized format. To ensure consistency in the integration process, we established an **agent integration protocol**. During enrolling, the developer completes a predefined form outlining the agent's capabilities, limitations, applications it interacts with, and demonstrations of its functionality (in Figure 2). Formally, the set of all enrolled agents is represented as $\mathcal{A} = \{(a_1, d_1), (a_2, d_2), ..., (a_n, d_n)\}$, where the completed form for each agent $a_i$ constitutes a document $d_i$. For specific examples of forms and documents, refer to the Appendix B.

**MetaAgent:** As the core of AgentStore, MetaAgent functions as the platform's manager. As shown on the right side in Figure 2, when a user provides a task, MetaAgent combines the task description with the system state (including screenshots, terminal output, accessibility tree, etc.) to select the appropriate agents from the AgentPool to complete it. This involves two primary functions. First, MetaAgent acts as a router, choosing the most suitable agent when a single agent can handle the task. Second, when multiple agents are required, MetaAgent divides the task into subtasks and assigns each to the appropriate agents, ensuring efficient task completion. In the next section, we will explain how MetaAgent performs inference to enable dynamic management.

## 3.2 METAAGENT WITH AGENTTOKEN

We employ the powerful open-source MLLM as the foundation for our MetaAgent $M$. This enables it to process multi-modal information covering task descriptions and OS states. Given the set of all enrolled agents $\mathcal{A}$, the goal of MetaAgent is to call a subset of these agents to automate computer tasks. Since the number of agents in AgentStore dynamically grows and reaches a large scale, common methods like In-Context Learning (ICL) (Chase, 2022; Li et al., 2023; Suzgun & Kalai, 2024) and full Fine-Tuning (FT) (Qin et al., 2023) become impractical due to the excessive context length and the high cost of retraining, respectively. Therefore, we propose the **AgentToken** strategy, which eliminates the need for lengthy contexts and significantly reduces the cost of retraining MetaAgent whenever a new agent is added.

Inspired by ToolkenGPT (Hao et al., 2024), which captures tool semantics using special tokens, AgentToken extends this concept by encoding enrolled agents as special tokens in the MetaAgent's vocabulary. Specifically, the agent tokens are parameterized as an embedding matrix $W_{\mathcal{A}} \in \mathbb{R}^{|\mathcal{A}| \times d}$ and appended to the original word token head $W_{\nu} \in \mathbb{R}^{|\mathcal{V}| \times d}$. Assuming the agent tokens $W_{\mathcal{A}}$ have been trained and available (as described in Section 3.3), the concatenated result forms the new language modeling head of MetaAgent. In this way, MetaAgent predicts the next token with the following probability:

$$P_M(t_i|t_{<i}) = \text{softmax}([W_{\nu}; W_{\mathcal{A}}] \cdot h_{i-1}),$$

where the next token can be either a word token or an agent token, *i.e.*, $t_i \in \mathcal{V} \cup \mathcal{A}$,. The operation $[;]$ denotes concatenation, and $h_{i-1} \in \mathbb{R}^d$ represents the last hidden state. In this context, AgentToken enables MetaAgent to fulfill its two primary functions:

**MetaAgent as Router**: Following the above manner, the most probable next token is obtained by maximizing the conditional probability:

$$t_i^* = \arg\max_{t \in \mathcal{V} \cup \mathcal{A}} \left( P_M(t_i|t_{<i}) \right).$$

Once an agent token is predicted, *i.e.*, $t_i^* \in \mathcal{A}$, the MetaAgent halts decoding, and the corresponding agent is invoked to execute the task. As illustrated in Figure 2, the above method enables MetaAgent to act as an efficient router, predicting the most appropriate agent to complete a task when a single agent is sufficient. However, many complex tasks require the collaboration of multiple agents. To address this, we extend the method by introducing a Manager mode.

**MetaAgent as Hash Manager**: We discover that, although each agent token is trained on individual tasks, they exhibit generalization capabilities for complex, collaborative tasks. Specifically, when a task requires multiple agents, the trained agent tokens often appear among the top candidates in the next token predictions. This observation led us to enhance this approach by shifting from single-token to multi-token prediction:

$$T_i^* = \text{TopK}_{t \in \mathcal{A}} \left( P_M(t_i | t_{<i}), \ K \right),$$

where $\text{TopK}(\cdot)$ is a function that returns the set of $K$ tokens from the vocabulary $\mathcal{A}$ that have the highest probabilities. These predicted tokens represent the $K$ agents most relevant to this task. The MetaAgent then switches to Manager mode by using a new prompt consisting of in-context documents for these selected agents, outlining how to generate subtasks for the complex task and assign them to the corresponding agents. Unlike previous methods that rely entirely on ICL for management, our method narrows the management scope to a few selected agents, leaving ample context space for detailed documentation of these fixed agents. This design shares similarities with hashing methods (Aggarwal & Verma, 2015), which convert inputs of arbitrary size into fixed-size outputs to facilitate retrieval and other operations. Therefore, we refer to this approach as *MetaAgent as Hash Manager*. It is important to note that the selection for the router and manager mode can be either manual or automatic. In the automatic setting, MetaAgent follows chain-of-thought (CoT; Wei et al., 2022), analyzing the given task to determine which mode to select and then switching to either router or manager. The base MetaAgent performs sufficiently well in making this binary decision without additional training.

### 3.3 TRAINING AGENTTOKEN WITH SELF-INSTRUCT

The embedding $W_{\mathcal{A}}$ corresponding to agent tokens are the only tunable parameters, introducing minimal additional training overhead. However, training these agent tokens requires a number of agent demonstrations that consist of the task descriptions and initial OS states. The corresponding token demonstrations were pre-collected for training in previous efforts (Hao et al., 2024; Chai et al., 2024). However, this strategy is not applicable in our scenario, as developers only provide a document about the agent, and it is unrealistic to expect them to supply massive demonstrations. Therefore, we propose an automated process with self-instruct (Wang et al., 2023c) for tuning these tokens using demonstrations from the MetaAgent itself.

The overall process follows an iterative algorithm to guide the generation of extra demonstrations, beginning with a limited set of original demonstrations $S_i = \{(y_k)\}_{k=1}^{n_i}$ and the agent description $c_i$ provided in document $d_i$. Specifically, we first prompt MetaAgent with existing demonstrations and agent descriptions:

$$S_i' = M(S_i, c_i),$$

where MetaAgent $M$ is expected to produce the new set of demonstrations $S_i'$. Following this, to ensure the quality of the generated outputs, we apply BERTScore (Zhang et al., 2019) to all newly generated outputs $y' \in S_i'$, ensuring both consistency and diversity. Specifically, we use a greedy algorithm (see Appendix C) to iteratively filter elements from $S_i'$, resulting in a refined set $S_i^{new} \subseteq S_i'$. The new set satisfies the following conditions:

$$\tau_1 \leq \text{BETRScore}(y_k, y_j) \leq \tau_2, \quad \forall y_k, y_j \in S_i \cup S_i^{new} \text{ and } k \neq j,$$

where $\text{BETRScore}(\cdot)$ represents the similarity between two demonstrations, with imposing a lower bound $\tau_1$ to avoid overly irrelevant outputs and $\tau_2$ ensuring diversity among them. In this way, we automatically filter the generated data, and the refined set is merged, *i.e.*, $S_i = S_i \cup S_i^{new}$.

The entire process is an automated iterative bootstrapping. MetaAgent further generates additional examples based on the augmented $S_i$, with BERTScore guiding and filtering the outputs until a sufficient number of demonstrations are generated to meet the training requirements for AgentToken.

**Training with self-generated data:** During training, each task description and initial state in demonstrations $S_i$ serve as the prefix, and a special agent token `<Agent_i>` is appended as the

ground truth for the next token prediction. Specifically, the training objective of AgentToken is:

$$\mathcal{L}(W_\mathcal{A}) = \sum_i^{|\mathcal{A}|} \sum_{y_j \in S_i} -\log P(\texttt{<Agent\_i>}|y_j),$$

the embedding $W_\mathcal{A}$ represents the only tunable parameters for all agents $\mathcal{A}$ in AgentPool. Notably, this training paradigm offers significant advantages in both efficiency and effectiveness. First, it eliminates the need for gradients to flow through the main body of MLLM parameters, resulting in more stable and efficient training than other efficient tuning methods (Hu et al., 2022; Lester et al., 2021). Second, AgentToken simply introduces additional tokens to the MetaAgent. The original language generation of the MLLM remains entirely unaffected as long as only the agent tokens are masked. This guarantees that the ICL method can be invoked seamlessly throughout the process.

Though inspired by (Hao et al., 2024), it diverges significantly in its application of token learning. First, previous methods are limited to single-modal and are not well-suited for handling multi-modal information in OS environments. Additionally, AgentToken extends token learning from single-token to multi-token prediction, enabling collaboration among multiple agents to automate complex tasks. Finally, due to the dynamic integration nature of our platform, we introduce automated iterative training with self-instruct, allowing continuous training of newly added agents without the need for pre-collected data, greatly enhancing the platform's scalability and flexibility.

## 4 EXPERIMENTS

To assess the effectiveness and versatility of **AgentStore**, we conducted comprehensive experiments across a diverse range of tasks. These experiments aimed to address two key questions: (1) **How crucial is the scalable integration of heterogeneous agents in AgentStore?** (2) **How important is AgentToken for dynamically managing a large number of agents in AgentStore?** In the following sections, we present our experimental results and offer a comparative analysis.

**Benchmark**   **OSWorld** (Xie et al., 2024) provides a scalable and real environment for evaluating computer agents, encompassing 369 tasks involving real web and desktop applications across open domains. As one of the most realistic and challenging benchmarks, OSWorld is ideal for capturing the diversity and complexity of real-world computer tasks, making it well-suited for testing the capability range of agents. Thus we selected OSWorld as the primary platform for our experiments. For more detailed information on OSWorld, please refer to the Appendix D.

**Settings**   We employ InternVL2-8B (Chen et al., 2024b) as the base model of our MetaAgent. Additionally, details regarding the Agents in the AgentPool can be found in Appendix A, along with the threshold selection for $\tau_1$ and $\tau_2$ in Appendix C. We generated about 100 examples for each agent using self-instruct for token training. The AdamW optimizer was used with a learning rate of 4e-5 and a weight decay of 1.0, for a total of 10 training epochs. When executing the Hash Manager, $K$ was set to 5. Further details on prompts can be found in the Appendix F.

### 4.1 HOW CRUCIAL IS THE SCALABLE INTEGRATION OF HETEROGENEOUS AGENTS?

#### 4.1.1 MAIN RESULTS ON OSWORLD

Table 1 presents the performance comparison between our approach and previous SoTA generalist agents on OSworld. While more advanced base models can improve performance (*e.g.*, GPT-4o outperforming GogVLM in **CogAgent** (Wang et al., 2023b; Hong et al., 2024)), even the best base models still face significant challenges. Notably, these methods exhibit not only overall weak performance but also significant disparities and weaknesses in specific task categories, despite using the same base models. For instance, **MMAgent** (Xie et al., 2024) and **CRADLE** (Tan et al., 2024) struggle with calculation tasks due to their lack of knowledge and operational capability in Excel, while **Friday** (Wu et al., 2024) and **Open-Interpreter** (ope, 2024), CLI-based agents, fails to execute GUI operation effectively in tasks, *e.g.*, Chrome or Thunderbird.

In contrast, AgentStore overcomes the limitations of previous methods by integrating over 20 specialized agents, each proficient in specific software and operations. "AgentStore(GT)" in Table

Table 1: Detailed success rates of previous methods and AgentStore on OSWorld, divided by apps (domains). Methods marked with "*" represent our re-implementation of the corresponding agents to ensure their applicability. Additionally, due to the significant overlap of operations between the OS and Workflow domains in the original division, we have merged these two domains into "OS*".

| Agent | Base | Success Rate (%) | | | | | | | | | |
|---|---|---|---|---|---|---|---|---|---|---|---|
| | | OS* | Calc | Impress | Writer | VLC | TB | Chrome | VSC | GIMP | AVG |
| CogAgent | GogVLM | 1.60 | 2.17 | 0.00 | 4.35 | 6.53 | 0.00 | 2.17 | 0.00 | 0.00 | 1.32 |
| MMAgent | GPT-4o | 14.44 | 4.26 | 6.81 | 8.70 | 9.50 | 6.67 | 15.22 | 30.43 | 0.00 | 11.21 |
| CRADLE | GPT-4o | 8.00 | 0.00 | 4.65 | 8.70 | 6.53 | 0.00 | 8.70 | 0.00 | 38.46 | 7.81 |
| Friday* | GPT-4o | 15.20 | 25.50 | 0.00 | 21.73 | 0.00 | 0.00 | 0.00 | 17.39 | 15.38 | 11.11 |
| Open-Inter* | GPT-4o | 12.80 | 12.76 | 0.00 | 13.04 | 0.00 | 0.00 | 0.00 | 17.39 | 15.38 | 8.94 |
| AgentStore(GT) | Hybrid | 20.00 | 36.17 | 10.63 | 47.83 | 47.06 | 40.00 | 34.78 | 47.82 | 38.46 | 29.54 |
| AgentStore(ICL) | Hybrid | 9.60 | 0.00 | 2.13 | 4.34 | 35.29 | 33.33 | 30.43 | 30.43 | 15.38 | 13.55 |
| AgentStore(FT) | Hybrid | 8.80 | 27.65 | 4.26 | 13.04 | 41.17 | 40.00 | 34.78 | 8.60 | 15.38 | 17.34 |
| AgentStore(AT) | Hybrid | 13.86 | 31.91 | 8.51 | 39.13 | 47.06 | 40.00 | 32.61 | 39.13 | 30.77 | 23.85 |

1 refers to each task being assigned to the most suitable agents, representing the upper bound of performance for the current AgentStore implementation. As shown, using specialized agents to handle tasks in their respective domains consistently outperforms generalist agents, with no significant performance shortcomings in almost all domains. This underscores the importance of various capabilities. Furthermore, when different methods are used to manage task allocation, all approaches outperform previous single-agent systems. AgentToken (AT) demonstrates the best performance due to its superior management abilities. We will elaborate on this in Section 4.2.

### 4.1.2 ANALYSIS OF AGENT QUANTITY AND DIVERSITY

To comprehensively analyze the advantages of scalable integration, we further explore the impact of the number and type of integrated agents within AgentStore on performance. To ensure thoroughness, we analyze AgentStore starting from a generalist MMAgent and incrementally add feature-specific agents in AgentPool to compare their effects on overall performance.

We employ two strategies for adding agents: one involves randomly selecting agents to incrementally add to the AgentPool, while the other categorizes agents into GUI and CLI types, starting with one type before supplementing with the other. As shown in Figure 3, performance gradually increases with the growing number of agents, confirming the performance benefits of scalable integration within AgentStore. Additionally, we observe differ-

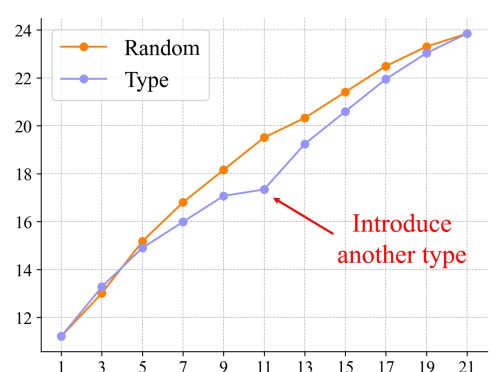

Figure 3: The performance curve as the number of agents increases, with the y-axis representing the success rate (%) on OSWorld and the horizontal x-axis representing the number of agents.

ences between the two strategies: random selection maintains a consistent mix of agent types, leading to a more stable growth. In contrast, adding agents of only one type causes the growth rate to slow over time, but this is mitigated when the other type is introduced. This highlights the crucial role of agent diversity, demonstrating the importance of integrating heterogeneous agents. These findings emphasize that both the quantity and diversity of agents are key factors in AgentStore.

### 4.1.3 GENERALIZATION ON MOBILE OS PLATFORMS

We further validate that AgentStore can generalize to mobile OS platforms. For this, we use the **APPAgent** (Yang et al., 2023) benchmark, which consists of nine popular mobile applications, each serving distinct purposes and collectively forming 45 tasks. Since the operations of mobile apps are entirely GUI-based, we design a dedicated agent for each app (a total of nine agents), which

differs from AgentStore in computer environments. Specifically, these agents are generated through a combination of self-exploration and human demonstrations within their respective applications.

Table 2 compares the performance of a single general agent with AgentStore on the APPAgent benchmark. As shown, the performance of the generalist agent, lacking specific knowledge of each app, is subpar across many applications, even when utilizing the strongest base model. In contrast, AgentStore constructs dedicated agents tailored to their respective applications, effectively addressing performance deficiencies in certain apps and demonstrating a significant performance improvement from 26.7% to 57.8%. This underscores the applicability of the AgentStore concept to other operating system platforms, highlighting its broader potential for application.

Table 2: Success rates of generalist agents and AgentStore. Methods marked with "*" indicate the re-implementation of the APPAgent without app-specific knowledge. *Due to differences between the original paper and the publicly available benchmark, the results may vary.* Additionally, while enhanced Appagent also generated app-specific agents, it did not integrate them into a complete system, instead only evaluating individual apps, and thus it is not included in the comparison.

| Agent | Base | Success Rate (%) | | | | | | | | | |
|---|---|---|---|---|---|---|---|---|---|---|---|
| | | Maps | X | TG | Temu | YT | Spotify | Yelp | Gmail | Clock | AVG |
| AppAgent* | Qwen-VL | 20.0 | 0.0 | 0.0 | 0.0 | 0.0 | 0.0 | 0.0 | 0.0 | 20.0 | 4.4 |
| AppAgent* | GPT-4o | 60.0 | 20.0 | 20.0 | 0.0 | 40.0 | 20.0 | 20.0 | 20.0 | 40.0 | 26.7 |
| AgentStore(GT) | GPT-4o | 80.0 | 60.0 | 40.0 | 40.0 | 60.0 | 80.0 | 80.0 | 60.0 | 60.0 | 66.7 |
| AgentStore(AT) | GPT-4o | 80.0 | 40.0 | 40.0 | 40.0 | 60.0 | 60.0 | 80.0 | 60.0 | 60.0 | 57.8 |

## 4.2 HOW IMPORTANT IS AGENTTOKEN FOR DYNAMICALLY MANAGING AGENTS?

In this section, extensive experiments demonstrate that AgentToken can enable MetaAgent to efficiently manage numerous agents, consistently outperforming advanced In-Context Learning (ICL) and Fine-Tuning (FT) techniques. We first evaluate MetaAgent's routing capability using the OS-World benchmark, demonstrating the advantages of the AgentToken strategy in terms of effectiveness, efficiency, and low data requirements. Additionally, we assess its collaborative management ability on a newly proposed multi-agent tasks benchmark.

### 4.2.1 METAAGENT AS ROUTER

Table 3: Routing success rates of different strategies for enabling MetaAgent as the router.

| Agent | Base | Success Rate (%) | | | | | | | | | |
|---|---|---|---|---|---|---|---|---|---|---|---|
| | | OS | Calc | Impress | Writer | VLC | TB | Chrome | VSC | GIMP | AVG |
| ICL | GPT-4o | 58.33 | 14.89 | 12.77 | 13.04 | 88.24 | 100 | 97.83 | 60.87 | 53.85 | 49.63 |
| ICL | InternVL | 37.50 | 6.38 | 21.28 | 8.70 | 35.29 | 33.33 | 52.17 | 30.43 | 30.77 | 41.57 |
| FT-LoRA | InternVL | 50.00 | 74.47 | 55.32 | 13.04 | 88.23 | 100 | 89.13 | 30.43 | 34.61 | 60.82 |
| AgentToken | InternVL | 75.00 | 80.85 | 72.34 | 43.47 | 100 | 100 | 95.65 | 91.30 | 73.08 | 80.60 |

**Effectiveness** As shown in Table 3, ICL methods perform poorly as routers, even when using advanced models like GPT-4o. This confirms our assertion that relying on simple descriptions and few-shot demonstrations to master new agents can be challenging. In contrast, other tuning methods show some improvement by training on more task demonstrations. However, these methods are highly dependent on the quantity of data (as discussed in the following sections), while their overall performance improvement remains marginal. In comparison, our AgentToken overcomes these challenges, requiring only minimal self-generated data to efficiently train the corresponding agent tokens. It demonstrates the most robust router capability. As shown in the bottom section of Table 1, after routing tasks through AgentToken, our AgentStore achieved a success rate of 23.85% on OSworld, significantly outperforming both ICL and FT strategies.

**Efficiency**   In Table 4, we compared the efficiency of the AgentToken with other efficient-tuning methods, *i.e.*, prompt tuning (Pt) and adapter tuning (LoRA), focusing on the number of trainable parameters, memory requirements, and training time on the same A100 device. Results indicate that AgentToken is the most efficient across all dimensions, requiring the least

Table 4: Efficiency comparison.

| Method | Params | Memory | Time |
|---|---|---|---|
| FT-Full | 7.78B | >80G | - |
| FT-Pt | 86K | 26G | - |
| FT-LoRA | 38M | 28G | 2.5 hours |
| **AgentToken** | **86K** | **17G** | **0.2 hours** |

amount of parameters and memory with the shortest training duration. Specifically, because Agent-Token eliminates the need for gradients to flow through the main body of MLLM, training time is significantly reduced, and the process becomes more stable. Conversely, full fine-tuning and prompt tuning suffer from instability due to their sensitivity to data, failing to converge properly.

**Data Requirement**   Generally, the larger and higher-quality the demonstration set $S_i$, the more beneficial it is for training AgentToken. However, in practical scenarios, manually acquiring a large volume of high-quality demonstrations poses significant challenges. The proposed automated process can mitigate this issue by generating data automatically; nevertheless, the scope of the generated data remains relatively limited (Shumailov et al., 2024). Consequently, previous tuning methods often experience reduced performance or even fail to converge. Fortunately, AgentToken can still be effectively trained due to its small parameter size and stable training process. As shown in Figure 4, when the demonstration set size reaches 100, a satisfactory accuracy rate can be achieved, aligning with prior methods (Hao et al., 2024; Chai et al., 2024). Based on this, we utilize a demonstration set size of 100 per agent in our experiments to train the tokens.

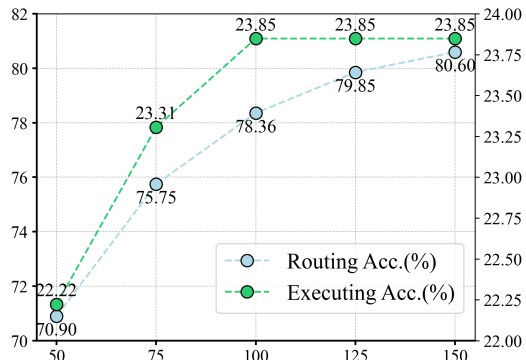

Figure 4: The accuracy curves with increasing training data corresponding to one agent. The x-axis represents the demonstration set size corresponding to each agent. The left y-axis represents the routing accuracy while the right y-axis indicates the executing accuracy.

### 4.2.2   METAAGENT AS HASH MANAGER

Although the existing OSWorld includes a limited number of tasks involving multi-agent collaboration, the small quantity and overly complex subtasks make it challenging to conduct meaningful experiments on collaborative task processing. Therefore, to further evaluate MetaAgent's ability to predict and coordinate multiple agents for collaborative task execution, we developed a new benchmark based on OS-World, comprising over 100 diverse tasks

Table 5: Performance comparison of collaborative task processing across different methods.

| Method | Base | Agent Match | Subtask Acc | Execution Acc |
|---|---|---|---|---|
| ICL | GPT-4o | 28.71% | 51.72% | 14.85% |
| ICL | InternVL | 24.75% | 40.00% | 9.90% |
| FT | InternVL | - | - | - |
| **AT** | **InternVL** | **36.63%** | **62.16%** | **22.77%** |

paired with agents in the AgentPool. This newly proposed benchmark allows us to assess the accuracy of both task decomposition and subtasks handling in a real environment. Additionally, we propose three metrics for evaluation: AgentMatch, SubtaskAcc, and ExecutionAcc, which respectively measure multi-agent prediction accuracy, subtask decomposition accuracy, and execution success rate. Detailed benchmark constructions and metric descriptions are provided in Appendix E.

As shown in Table 5, the FT method is not applicable in this scenario due to the infinite combinations of agents, making it impossible to pre-organize the necessary data for training. Moreover, while the ICL methods function to a certain extent, even with advanced commercial models, the constraints of overly long contexts and vast combinatorial spaces result in subpar outcomes. In contrast, AgentToken leverages its inherent task awareness, employing a hashing mechanism to significantly narrow the scope to a few selected agents, thereby demonstrating excellent performance across all metrics.

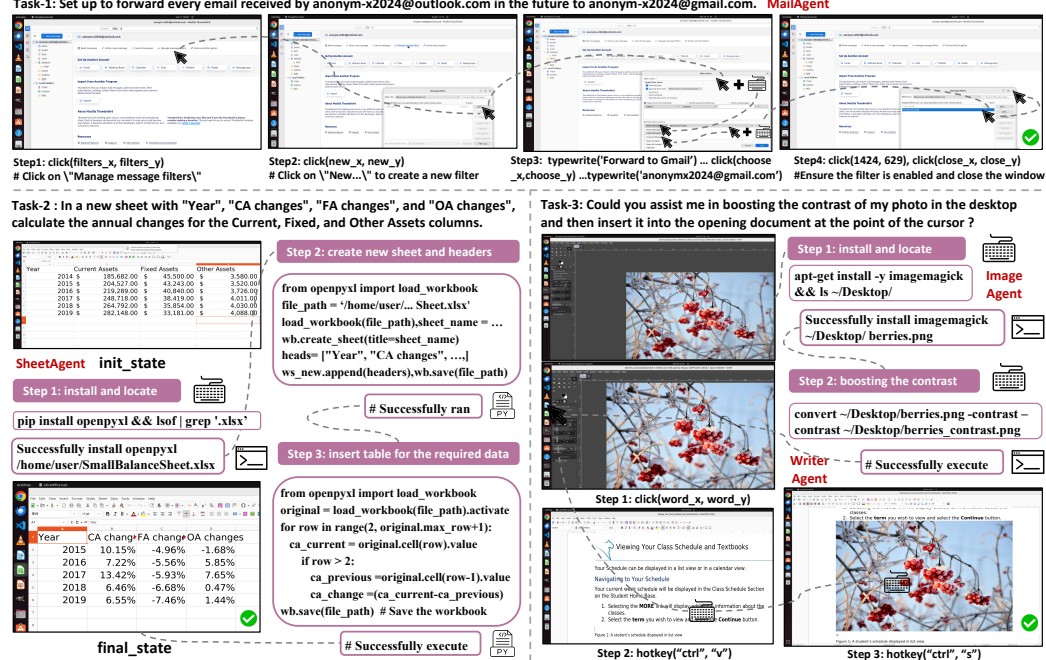

Figure 5: Specific steps involved in executing three tasks mentioned in the qualitative analysis.

## 4.3 QUALITATIVE ANALYSIS

In Figure 5, we highlight representative examples of outcomes, along with detailed analysis, to illustrate how AgentStore enhances the overall system's capability to tackle complex, open-ended tasks in real-world environments. In Task-1, the agent is tasked with setting up automatic email forwarding, which involves frequent GUI interactions and requires a strong understanding of Thunderbird's layout and forwarding settings, posing challenges for those unfamiliar with email systems. However, when MetaAgent assigns the specialized MailAgent to handle the task, the agent efficiently navigates the software, knowing the exact steps to configure the forwarding settings. In particular, during the Step3, it executes a sequence of actions to accurately fill out the required forms and options, showcasing its advanced understanding and processing capabilities within the mail domain. Similarly, in Example 2, which requires complex processing of a spreadsheet, MetaAgent selects the SheetAgent from the AgentPool to handle the task, avoiding overly complex GUI interactions. SheetAgent possesses knowledge of "openpyxl" and a deep understanding of the steps needed to manipulate sheets, efficiently completing this task that is too challenging for previous generalist agents (Xie et al., 2024; Tan et al., 2024). In addition, Example 3 illustrates a system-wide task that requires collaboration among multiple agents. MetaAgent successfully decomposes the task into subtasks and assigns the appropriate agents to complete each one. This demonstrates AgentStore's ability to perceive the overall task structure, overcoming the limitations of isolated, single-specialist agents and showcasing its strong generalization capability. In summary, these examples highlight AgentStore's specialized generalist abilities in handling not only domain-specific but also system-wide tasks, underscoring its potential for building a specialized generalist computer assistant.

## 5 CONCLUSION

In this paper, we introduce AgentStore, a flexible and scalable platform for dynamically integrating various heterogeneous agents to independently or collaboratively complete complex OS tasks. Furthermore, we propose MetaAgent with the AgentToken strategy to achieve efficient management of the growing number of agents. Extensive experimental results validate both the importance of scalable integration and the effectiveness of the AgentToken strategy. Comprehensive quantitative analysis and qualitative results show that AgentStore expands the capabilities of existing agent systems in both generalization and specialization. We believe that as basic AGI models continue to evolve, AgentStore, as an open platform, will integrate more powerful agents, progressively advancing toward the vision of building the specialized generalist computer assistant.

ETHICS STATEMENT

This research focuses on building a scalable platform to integrate heterogeneous agents dynamically. The data datasets or benchmarks we employed are properly cited. There are no discrimination, bias, or fairness issues that need to be declared in this paper. Further, the outputs are not expected to be potentially harmful. To ensure reproducibility, we provide all experimental details in Section 4 and their corresponding appendices. All source code will be made public.

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

# A   AGENTPOOL

The AgentPool is a collection of all available agents within AgentStore. To build the prototype of AgentStore, we organized 20 agents within Agent-Pool, each with distinct functionalities. As shown in Table 6, these agents range from unimodal to multimodal, from open-source to closed-source models, and from Command-Line Interfaces (CLI) to Graphical User Interfaces (GUI). The diverse capabilities of these agents cover common applications and tasks in both daily life and professional settings. In addition to the domain-specific agents we developed, we also integrated existing agents, such as Friday (Wu et al., 2024) and (He et al., 2024). This demonstrates the scalability of our approach, which allows third-party agents to be added to the platform.

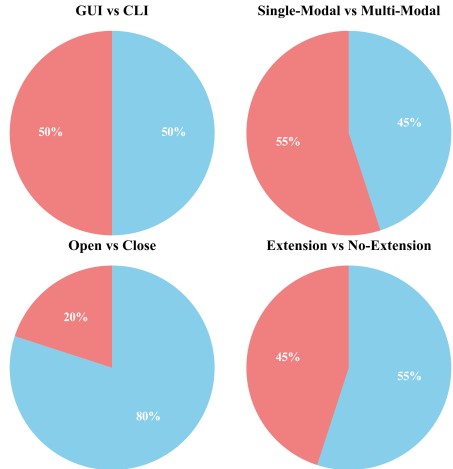

Figure 6: The agent distribution across different types.

Specifically, for closed-source model agents, we uniformly use GPT-4o as the base model. For open-source model agents, single-modality agents are based on Llama 3.1 (Touvron et al., 2023), while multi-modality agents are built on InternVL2 (Chen et al., 2024b). The last column of Table 6 indicates whether the agent has the capability to solve tasks outside its own domain.

Figure 6 illustrates the distribution of different types of agents, showing that the initial version of AgentStore maintains a consistent balance between GUI and CLI agents. Most models also support extensions to handle additional tasks. Due to the significant gap between open-source and close-commercial models, most agents in this version are currently based on close-commercial models.

Table 6: The presentation of agents in the AgentPool.

|  | CLI or GUI? | Single or Multi Modal? | Open or Close Base Model? | Domain for OSworld | Support Extension? |
|---|---|---|---|---|---|
| OSAgent | GUI | Multi | Close | OS | ✓ |
| Friday (Wu et al., 2024) | CLI | Single | Close | OS | ✓ |
| SheetAgent | CLI | Single | Close | Calc | ✗ |
| CalcAgent | GUI | Multi | Close | Calc | ✓ |
| SlideAgent | CLI | Single | Close | Impress | ✗ |
| ImPressAgent | GUI | Multi | Close | Impress | ✓ |
| WordAgent | CLI | Single | Close | Writer | ✗ |
| WriterAgent | GUI | Multi | Close | Writer | ✓ |
| VLCAgent | GUI | Multi | Close | VLC | ✓ |
| MailAgent | GUI | Multi | Close | TB | ✓ |
| ChromeAgent | GUI | Multi | Close | Chrome | ✓ |
| WebAgent (He et al., 2024) | GUI | Multi | Close | Chrome | ✗ |
| VSAgent | GUI | Multi | Open | VSC | ✗ |
| VSGUIAgent | CLI | Single | Close | VSC | ✓ |
| GimpAgent | GUI | Multi | Close | GIMP | ✓ |
| ImageAgent | CLI | Single | Open | GIMP | ✓ |
| Searcher | CLI | Single | Close | - | ✗ |
| GoogleDrive | CLI | Single | Close | - | ✗ |
| CoderAgent | CLI | Single | Open | - | ✗ |
| VisionAgent | CLI | Multi | Open | - | ✗ |

# B AGENTENROLL

When a developer creates a new OS agent and seeks to integrate it into AgentStore, it is essential to register the agent's information in a standardized format. To ensure consistency in the integration process, we established an **agent integration protocol**. As shown in the template below, during enrollment, the developer completes a predefined form outlining the agent's capabilities, limitations, the applications it interacts with, and demonstrations of its functionality.

The completed form for each agent constitutes a document. Following the template, we present six typical agent documents related to LibreOffice tasks to help readers understand the AgentEnroll process and outcomes, as well as to provide a clearer view of the agents in the AgentPool. Due to space limitations, further details on additional agents will be available when the entire project is open-sourced.

In the actual enrollment process, we encourage developers to provide more demonstrations—the greater the number, the more comprehensive the document will be, which also facilitates agentToken training during the self-instruct process. In this paper, we provide 10 demonstrations for each agent, which is relatively lightweight but still effectively aids the Metaagent in learning and understanding the corresponding agent.

```
Templete: AgentName

# Applications:
# List the applications or tools that the agent supports
or interacts with.

# Capabilities
# Describe the main functions and abilities of the agent.
Include details about the tasks it can perform and the
libraries or technologies it utilizes.

# Limitations
# Outline the constraints and tasks the agent cannot perform.
This helps set clear boundaries for the agent's functionality.

# Demonstrations

# Demostation_1: <Description of the first demonstration task.>

# Demostation_2: <Description of the second demonstration task.>

# Demostation_3: <Description of the third demonstration task.>

# Demostation_4: <Description of the fourth demonstration task.>

......

End!
```

AgentName: *SlideAgent*

**# Applications:**
`Terminal,LibreOffice Impress`

**# Capabilities**
`Specializes in creating and modifying PowerPoint presentations using`
`Python's python-pptx library. It can handle tasks involving slide`
`creation, layout management, text and content insertion, and`
`formatting adjustments. Also capable of detecting open PowerPoint`
`presentations using Bash commands.`

**# Limitations**
`Cannot handle GUI operations, cannot perform tasks outside the`
`capabilities of the python-pptx library such as directly interacting`
`with embedded videos and complex animations. Additionally, cannot`
`modify LibreOffice Impress software defaults or preferences.`

**# Demostations**

`Demostation_1: Can you add a new slide at the end of my presentation`
`with the title 'Conclusion' and the text 'Thank you for your`
`attention'?"`

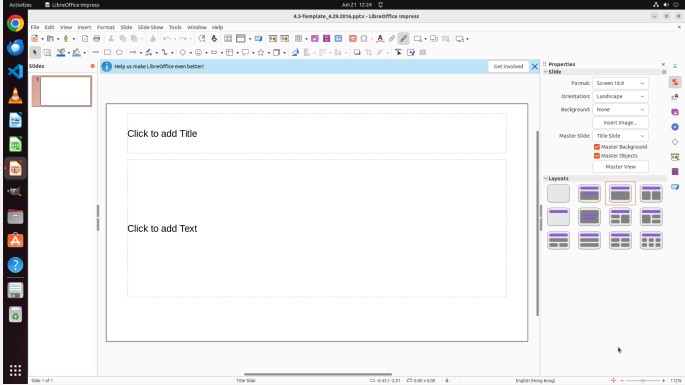

`Demostation_2: Can you add a footer with text 'Company Confidential'`
`to all slides in the current PowerPoint presentation?`

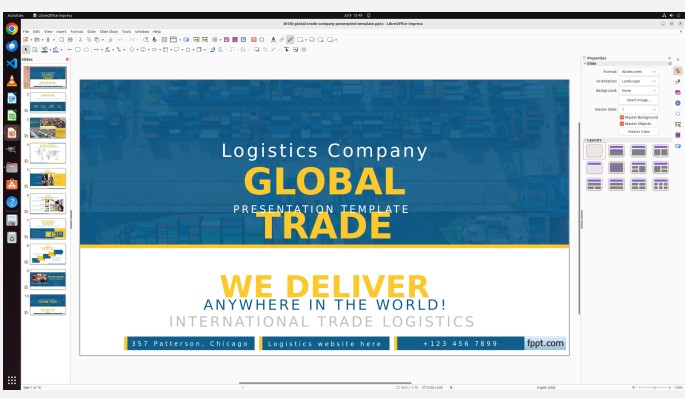

`......`

**End!**

AgentName: *ImPressAgent*

# Applications:
LibreOffice Impress

# Capabilities
Specializes in handling tasks using GUI operations and can modify
LibreOffice Impress software defaults or preferences.

# Limitations
Cannot handle complex tasks such as creating and modifying PowerPoint
presentations using Python's python-pptx library.

# Demonstrations

Demostation_1: Enable the "Grid" view to help with precise
placement of objects.

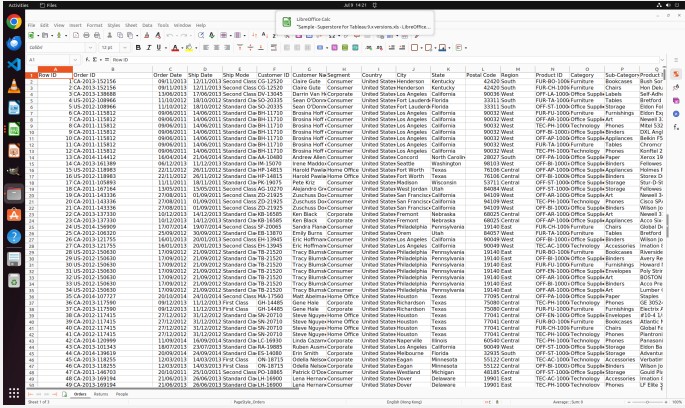

Demostation_2: Change the default font for all text in the
presentation to "Helvetica".

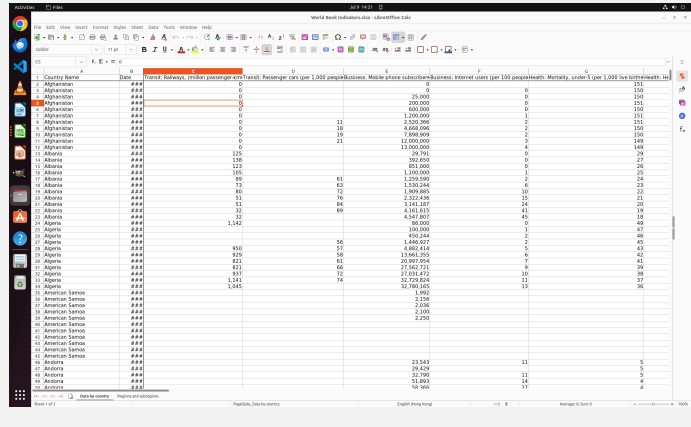

......

End!

AgentName: *WordAgent*

**# Applications:**
**Terminal, LibreOffice Writer**

**# Capabilities**
**Excels at identifying and manipulating Word documents using Python's python-docx library. Can manage tasks involving document modification, data insertion, and formatting adjustments. Capable of detecting open Word or other documents using Bash commands.**

**# Limitations**
**Cannot handle GUI operations, cannot perform tasks outside the capabilities of the python-docx library such as directly interacting with embedded media and scripts within the documents. Additionally, cannot modify LibreOffice Writer software defaults or preferences.**

**# Demonstrations**

**Demostation_1: Add the text 'Grand Opening' as a title at the beginning of the document."**

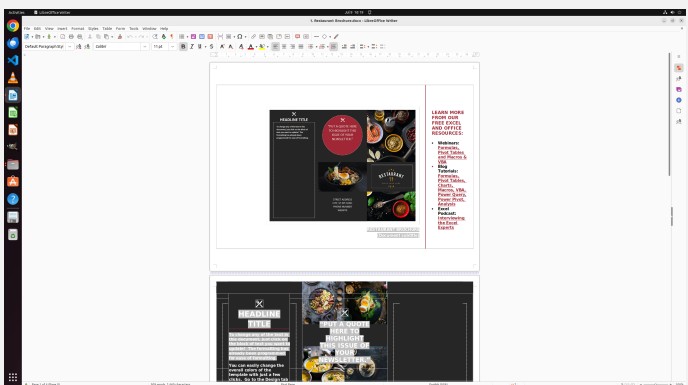

**Demostation_2: Insert a horizontal line above the 'ABSTRACT' heading.**

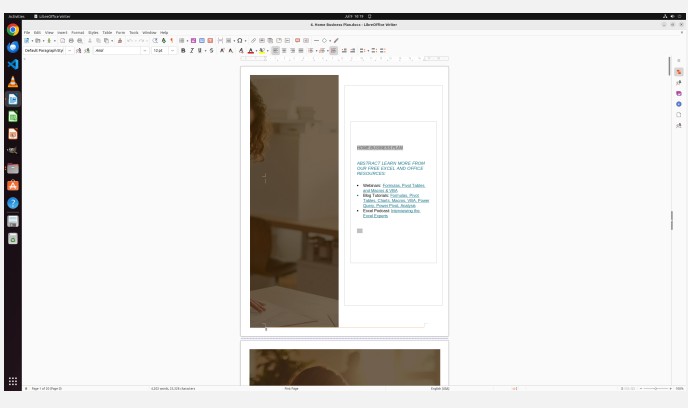

**. . . . . .**

**End!**

**AgentName:** *WriterAgent*

**# Applications:**
**LibreOffice Writer**

**# Capabilities**
**Specializes in handling GUI operations and can perform tasks outside the capabilities of the python-docx library, such as directly interacting with embedded media and scripts within documents. Can modify LibreOffice Writer  software defaults or preferences.**

**# Limitations**
**Cannot identify and manipulate Word documents using Python's python-docx library, and cannot manage tasks involving document modification, data insertion, and formatting adjustments. Additionally, cannot detect open  Word or other documents using Bash commands.**

**# Demonstrations**

**Demostation_1: Enable the "Show Changes" feature to track document edits location.**

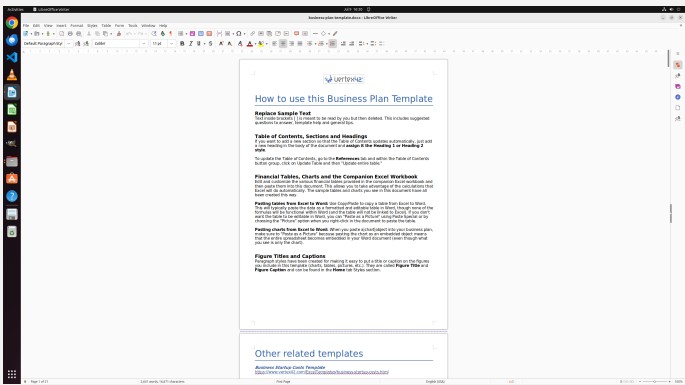

**Demostation_2: Create a custom keyboard shortcut for "Print" set to Ctrl+P.**

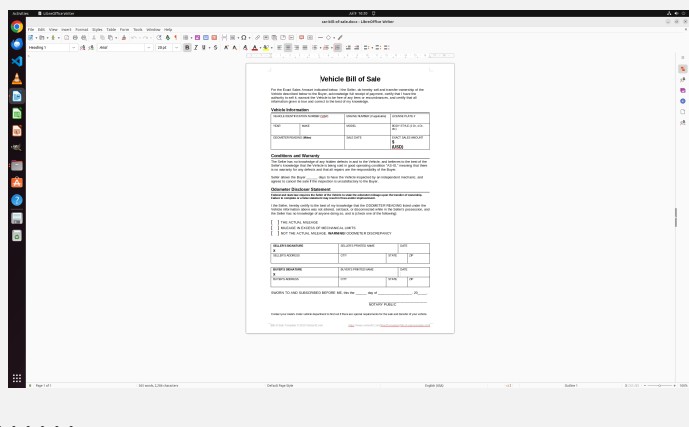

**......**

**End!**

AgentName: *SheetAgent*

# Applications:
Terminal, LibreOffice Calc

# Capabilities
Specializes in creating, analyzing, and modifying Excel spreadsheets using Python's openpyxl library. It can handle tasks involving data entry, formula insertion, chart creation, and spreadsheet formatting. Also capable of detecting open Excel files using Bash commands.

# Limitations
Cannot handle GUI operations, cannot perform tasks outside the capabilities of the openpyxl library such as directly interacting with complex macros. Additionally, cannot modify LibreOffice Calc software defaults or preferences.

# Demonstrations

Demostation_1: Highlight rows where the total sales exceed $1000.

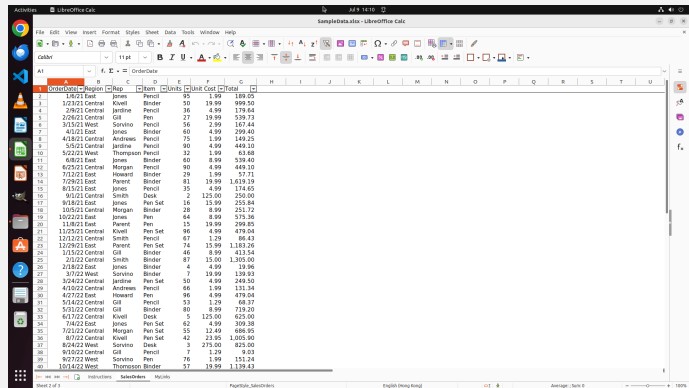

Demostation_2: Filter out players older than 35 and list their names and ages in a new sheet named "Veteran Players".

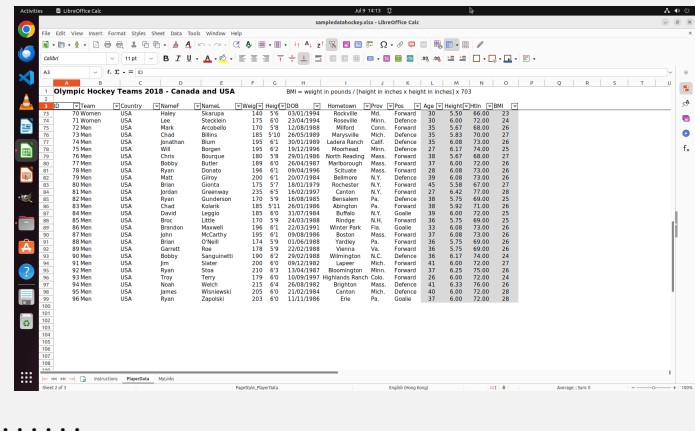

......

End!

**AgentName:** *CalcGUI*

**# Applications:**
LibreOffice Calc

**# Capabilities**
Specializes in handling tasks using GUI operations and can modify
LibreOffice Calc software defaults or preferences.

**# Limitations**
Cannot handle complex tasks such as creating, analyzing, and modifying
Excel spreadsheets using Python's openpyxl library.

**# Demonstrations**

Demostation_1: Set the row height to 18 pixels for better readability.

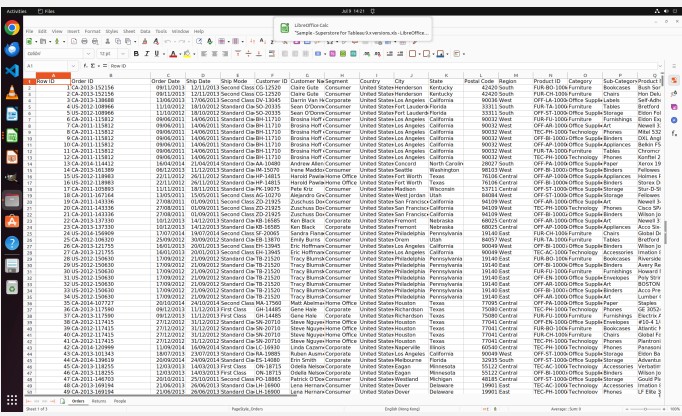

Demostation_2: Filter the data to show only rows where "Health:
Mortality, under-5" is greater than 50.

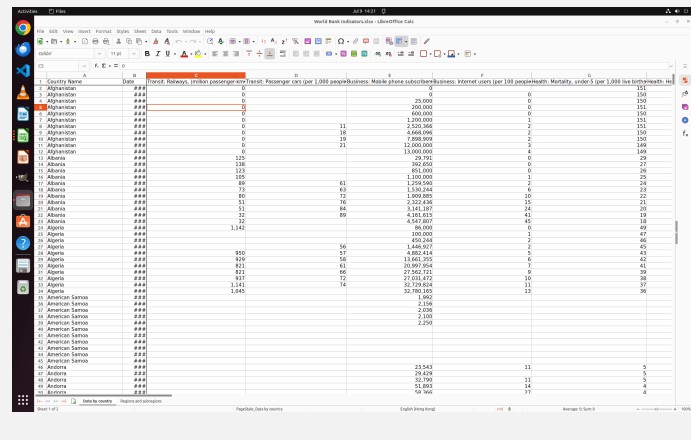

......

**End!**

## C AUTOMATED PROCESS WITH SELF-INSTRUCT

In this section, we provide more details about the Automated data generation process, including threshold selection and the greedy filtering algorithm.

**Threshold Selection**   To ensure the reliability of threshold selection, we first studied the distribution of thresholds in real-world tasks based on human-labeled standards. As shown in Figure 7, in tasks labeled by OSworld, the 95% threshold distribution of BertScore across different domains is primarily concentrated between 0.77 and 0.92. Therefore, to further strictly control the quality of generated data, we ultimately selected a threshold of 0.8 for $\tau_1$ and 0.9 for $\tau_2$ to filter the data.

This approach offers several advantages. By selecting thresholds of 0.8 for $\tau_1$ and 0.9 for $\tau_2$, we strike a balance between retaining high-quality data and ensuring the diversity necessary for robust training. The $\tau_1$ threshold helps in eliminating low-quality samples, while $\tau_2$ enforces stricter criteria for the final selection of data, ensuring that only the most relevant and high-quality data points are used. This dual-threshold filtering process not only improves the precision of the generated data but also enhances the overall performance of agent training, reducing the risk of overfitting to noise or irrelevant tasks.

**Greedy Filtering Algorithm**   Algorithm 1 presents a greedy algorithm for filtering a set of newly generated demonstrations, $S_i'$, ensuring that each selected demonstration maintains a BERTScore similarity within the specified bounds $\tau_1$ and $\tau_2$ relative to both existing demonstrations $S_i$ and previously selected new demonstrations $S_i^{new}$. The key improvement lies in the prioritization of demonstrations that are optimally positioned between the two thresholds, thereby enhancing both relevance and diversity.

A prioritization mechanism selects demonstrations optimally positioned between the similarity thresholds. By calculating the minimum distance of each candidate's BERTScore to the thresholds, the algorithm ensures that selected demonstrations are neither too similar nor too dissimilar to existing ones. This strategic ordering facilitates the inclusion of the most appropriate demonstrations first, thereby maximizing both the relevance and diversity of the refined set $S_i^{new}$. Consequently, the quality of the training data for AgentToken is significantly improved, fostering more effective training outcomes.

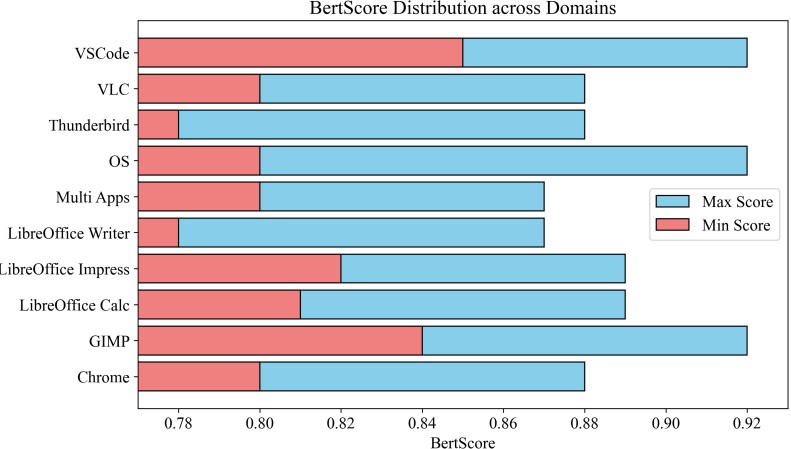

Figure 7: BertScore distribution across different domains.

---

**Algorithm 1** Greedy Filtering of Generated Demonstrations using BERTScore with Prioritized Selection

---

**Require:** • $S_i' = \{y_1', y_2', \ldots, y_m'\}$: Set of newly generated demonstrations
   • $S_i = \{y_1, y_2, \ldots, y_n\}$: Existing set of demonstrations
   • $\tau_1$: Lower bound for BERTScore similarity
   • $\tau_2$: Upper bound for BERTScore similarity
**Ensure:** • $S_i^{new}$: Refined set of new demonstrations satisfying the similarity constraints
 1: Initialize $S_i^{new} \leftarrow \emptyset$
 2: For each $y' \in S_i'$, compute the minimum distance to the thresholds:

$$d(y') = \min(|\text{BERTScore}(y', y) - \tau_1|, |\text{BERTScore}(y', y) - \tau_2|) \quad \forall y \in S_i$$

 3: Sort $S_i'$ in descending order based on $d(y')$
 4: **for** each $y' \in S_i'$ in sorted order **do**
 5:     Initialize a flag $valid \leftarrow$ True
 6:     **for** each $y \in S_i \cup S_i^{new}$ **do**
 7:         Compute BERTScore$(y', y)$
 8:         **if** BERTScore$(y', y) < \tau_1$ **or** BERTScore$(y', y) > \tau_2$ **then**
 9:             $valid \leftarrow$ False
10:             **break**
11:         **end if**
12:     **end for**
13:     **if** $valid$ **then**
14:         Add $y'$ to $S_i^{new}$
15:     **end if**
16: **end for**
17: **return** $S_i^{new}$

---

# D OSWORLD

OSWorld (Xie et al., 2024) is a scalable, computer environment designed for multimodal agents. This platform provides a real-world environment for assessing open-ended computer tasks involving various applications. In this section, we provide a detailed introduction to OSworld, focusing on three key aspects: the open-ended and diverse nature of tasks, the reliability of evaluations in real-world environments, and the varied capability requirements for agents. This aims to help readers understand the rationale behind using OSworld as the primary evaluation platform in the main text.

## D.1 OSWORLD TASKS

OSWorld is a benchmark suite consisting of 369 real-world computer tasks, primarily based in an Ubuntu Linux environment, with a smaller portion covering Microsoft Windows. The tasks are sourced from the authors as well as various platforms like forums, tutorials, and guidelines. Each task is paired with a natural language instruction and a hand-crafted evaluation script for scoring. Figure 8 provides a detailed classification of tasks, showcasing their diversity and effectively reflecting the nature of open-ended tasks in real-world scenarios.

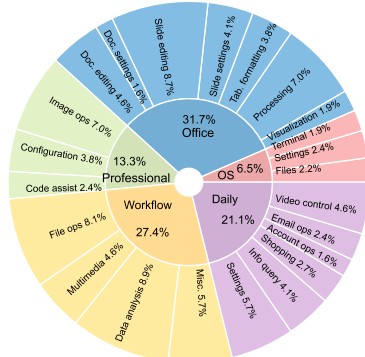

Figure 8: Task instructions distribution in OSWorld (Xie et al., 2024)

## D.2 REAL-WORLD COMPUTER ENVIRONMENT

As shown in Figure 9, OSworld provides an executable and controllable environment that supports task initialization, execution-based evaluation,

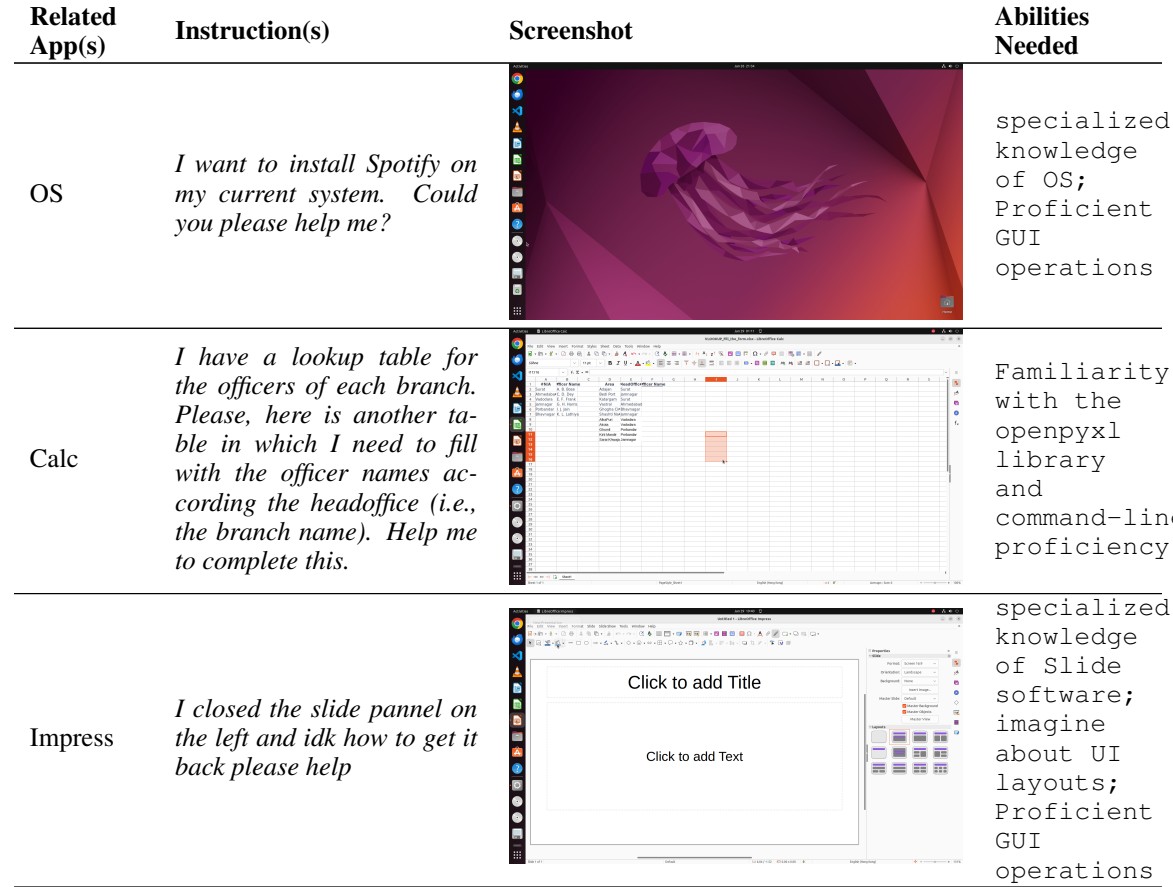

Figure 9: OSWorld can serve as a unified environment for evaluating *open-ended* computer tasks in the real-world computer environment.

and interactive agent learning in a range of *real* operating systems. It also provides easily accessible system screenshots, ally-tree information, and interfaces that facilitate agent output for mouse and keyboard operations. This rich system information, real-time execution, and comprehensive task evaluation offer an interactive environment that is not limited to specific applications or domains.

## D.3 REPRESENTITIVE EXAMPLES

In Table 7, we present several representative examples from OSworld, which aim to illustrate the distinct operational logic involved in different tasks and the diverse capabilities required. These examples help readers better understand the broad range of generalization and specialized skills necessary in real-world computer environments, which are challenging for a single agent to fully encompass.

Table 7: Representitive Examples from OSWorld to illustrate the distinct operational logic and the diverse capabilities involved in different tasks.

| Related App(s) | Instruction(s) | Screenshot | Abilities Needed |
|---|---|---|---|
| OS | *I want to install Spotify on my current system. Could you please help me?* | | `specialized knowledge of OS; Proficient GUI operations` |
| Calc | *I have a lookup table for the officers of each branch. Please, here is another table in which I need to fill with the officer names according the headoffice (i.e., the branch name). Help me to complete this.* | | `Familiarity with the openpyxl library and command-line proficiency` |
| Impress | *I closed the slide pannel on the left and idk how to get it back please help* | | `specialized knowledge of Slide software; imagine about UI layouts; Proficient GUI operations` |

**Table 7 – continued from previous page**

| Related App(s) | Task Instruction | Screenshot of Initial State | Abilities Needed |
|---|---|---|---|
| Chrome | *Can you help me clean up my computer by getting rid of all the tracking things that Amazon might have saved? I want to make sure my browsing is private and those sites don't remember me.* |  | `specialized knowledge of Chrome browser, proficient GUI operations` |
| VLC | *Hey, could you turn this video the right way up for me? And once it's flipped around, could you save it for me with the name '1984_Apple.mp4' on the main screen where all my files are?* |  | `software knowledge; spatial judgment ability` |
| Thunderbird | *Create a local folder called "Promotions" and create a filter to auto move the inbox emails whose subject contains "discount" to the new folder* |  | `Knowledge of the Thunderbird mail system; GUI operations` |
| VS Code | *Please modify VS Code's settings to disable error reporting for Python missing imports.* |  | `software knowledge to deal with settings; reasoning to understand the cause and solution of the error` |
| GIMP | *Could you tone down the brightness of my photo?* |  | `Proficiency in using ImageMagick and CLI operations` |

*Continued on next page*

**Table 7 – continued from previous page**

| Related App(s) | Task Instruction | Screenshot of Initial State | Abilities Needed |
|---|---|---|---|
| GIMP | *Help me choose the yellow triangle and position it at the center of my picture.* |  | `spatial perception and reasoning, as well as precise control of actions` |
| Multiple (VLC+GIMP) | *Could you help me create an Animated GIF from a video file using VLC and GIMP from the source of video "src.mp4", 5-second clip beginning at 00:03?* |  | `specialized software knowledge; generalization ability to process multi-step procedure successfully` |
| Multiple (Chrome+Calc) | *Could you help me extract data in the table from a new invoice uploaded to my Google Drive, then export it to a Libreoffice calc .xlsx file in the desktop?* |  | `specialized ability to do table data;generalization ability to process multi-step procedure successfully` |

## E   OSWORLD-MULTI BENCHMARK

Building on OSworld, we further developed a new benchmark, **OSWorld-Multi**, to evaluate MetaAgent's ability to predict and coordinate multiple agents for collaborative task execution. OSWorld-Multi consists of 101 tasks, each requiring collaboration with paired agents from the AgentPool. In the following sections, we will introduce the construction process, task examples, and evaluation metrics.

**Construction process**   To maximize the reuse of tasks, system states, and evaluation functions from OSworld, we adopted a reverse synthesis approach. By mining paired examples in OSworld, we generated tasks requiring agent collaboration. Specifically, we first traversed all pairwise combinations of subtasks, applying a two-step validation process: an initial filtering with a large language model (LLM), followed by manual review. This method allowed us to select meaningful collaborative tasks. Moreover, this approach enabled the synthesis of tasks requiring not only two-agent collaboration but also those involving three or more agents. In the following section, we will present some of the generated collaborative task results to demonstrate the outcomes of this synthesis process.

**Task examples**   In the table below, we present several synthesized examples to help readers understand the generation process. Another advantage of this reverse synthesis approach is the presence of natural ground truth, allowing us to evaluate not only execution accuracy but also the accuracy of agent predictions and task decomposition. This enables a comprehensive assessment of collab-

orative task execution. In the following sections, we will provide a detailed explanation of the corresponding evaluation metrics.

---

**Synthesis task 1**

```
# Agent:Subtask-1

VLCAgent:Snap a photo of the current video scene, save it as
'interstellar.png', and put it on the Desktop, please.

# Agent:Subtask-2

WriterAgent: Add page number for every page at the bottom left.

# Synthesis task

Capture a scene from a video in VLC and insert the image
into a LibreOffice document with a page number.

# Required:  VLCAgent + WriterAgent
```

---

**Synthesis task 2**

```
# Agent:Subtask-1

VLCAgent: Help me modify the folder used to store my
recordings to Desktop.

# Agent:Subtask-2

Friday: Change the permission of all regular files under
current directory tree to 644.

# Synthesis task

Modify VLC's recording folder to Desktop and set file
permissions to 644 for all files in this directory.

# Required: VLCAgent + Friday
```

---

**Synthesis task 3**

```
# Agent:Subtask-1

VLCAgent: Can you start streaming the video from this link for me?
https://www.youtube.com/watch?v=pgBsyTKAwLw

# Agent:Subtask-2

ChromeGUI: Could you help me clear browsing history from Youtube?

# Synthesis task

Could you stream a video from a YouTube link in VLC and clear
all YouTube browsing history in Chrome after to ensure a clean search
experience?

# Required: VLCAgent + ChromeGUI
```

> Synthesis task ......
>
> . . . . . .

**Evaluation metrics**    We propose three metrics for evaluation: AgentMatch, SubtaskAcc, and ExecutionAcc, which respectively measure multi-agent prediction accuracy, subtask decomposition accuracy, and execution success rate.

**AgentMatch** is designed to assess the accuracy of the agent prediction process during collaborative task execution. It compares the predicted set of agents selected by the MetaAgent with the ground truth set of agents that are required for successful task completion. Essentially, AgentMatch measures how well the MetaAgent can correctly identify the appropriate agents from the AgentPool for a given task. The metric is computed by calculating the accuracy of the predicted agent set relative to the actual agents involved in the task. Specifically, it checks whether the predicted agents match the expected agents. A high AgentMatch score indicates that the MetaAgent is effectively coordinating and predicting the correct agents for task execution.

**SubtaskAcc** is an evaluation metric that measures the accuracy of task decomposition by comparing the predicted subtasks assigned to each agent with the ground truth subtasks. It evaluates how well the MetaAgent decomposes a given task and assigns the correct subtasks to the respective agents. To assess SubtaskAcc, we use a textual comparison between the predicted subtasks and the actual subtasks for the same agent. This comparison is based on textual similarity, using BERTScore as the evaluation metric. As per our analysis in C, if the BERTScore is below 0.77, the two subtasks are considered too dissimilar, and the decomposition is deemed unsuccessful. Conversely, if the BERTScore exceeds this threshold, the decomposition is considered accurate. This threshold ensures that only decompositions with sufficiently high textual similarity are counted as correct. SubtaskAcc thus reflects how effectively the MetaAgent can break down a complex task and allocate the correct components to individual agents. A high SubtaskAcc score indicates that the MetaAgent is accurately identifying the required subtasks for each agent, contributing to the overall success of the collaborative task execution.

**ExecutionAcc** is an evaluation metric that measures the success rate of task execution by reusing the assessment methods from OSworld. This metric focuses on determining whether the predicted subtasks are correctly executed by the agents, based on their final state in the environment.

To evaluate ExecutionAcc, we rely on OSworld's system of getter and evaluator functions. The getter function extracts key components from the final state of the environment (e.g., a modified file or text contents displayed in a window element), while the evaluator function assesses success based on these extracted components. If a necessary function does not exist, it is constructed and added to the function library of the environment. Each task is evaluated by comparing its final execution state with the expected outcome, and the evaluation process is designed to be robust.

In the context of our system, ExecutionAcc provides a direct measure of how successfully the agents complete their assigned tasks, reflecting the practical performance of task execution in real-world scenarios. A high ExecutionAcc indicates that the agents are accurately following the predicted subtasks and completing them correctly in the environment.

# F    PROMPT DETAILS

We provide examples of MetaAgent prompts in different modes to help readers understand the inference process. It is important to note that in manager mode, the prompt templates in Section F.3 for AgentToken and ICL are identical. The key difference is that AgentToken reduces the number of input documents, effectively shortening the context length, which in turn improves performance.

Additional prompts, including those related to each individual agent and self-instruct, will be provided when the project is open-sourced.

## F.1 PROMPT FOR ROUTER MODE FOR AGENTTOKEN

> **Prompt:** *Router for AgentToken*
>
> ```
> Imagine you have a complex task that needs to be executed on an
> operating system.
> This task can be decomposed into sub-tasks corresponding to
> the model's capabilities.
> You have several agents with different specializations available.
> Requirements:
> The task is assigned to one agent, the model should return
> the one token of that agent.
> Now your task is {task_name}
> ```

## F.2 PROMPT FOR ROUTER MODE FOR ICL

> **Prompt:** *Router for ICL*
>
> ```
> Imagine you have a complex task that needs to be executed on an
> operating system.
> This task can be decomposed into sub-tasks corresponding to
> the model's capabilities.
> You have several agents with different specializations available.
> {agent_1_document},{agent_2_document},...{agent_n_document}
> Requirements:
> The task is assigned to one agent, the model should return the
> name of that agent.
> like:
> ###CalcAgent###
> Now your task is {task_name}
> ```

## F.3 PROMPT FOR MANAGER MODE

> **Prompt:** *Manager Mode*
>
> ```
> Imagine you have a complex task that needs to be executed
> on an operating system.
> This task can be decomposed into sub-tasks corresponding
> to the model's capabilities.
> You have agents with different specializations available:
> {agent_1_document},{agent_2_document},...{agent_n_document}
>
> Requirements:
> The task requires multiple agents, the model should specify
> which sub-tasks each agent should handle.
> The model should ensure that the task assignment optimizes
> efficiency and effectiveness, considering the unique
> capabilities of each agent.
> return like:
> ###AgentName1:compute the sum of data in a new sheet.###
> ###AgentName2:upload the computed file to the google Drive###
>
> Be careful not to assign the same agent to perform tasks
> consecutively.
> don't return like this:
> ###Agent1:compute the sum of data in a new sheet.###
> ###Agent1:rename this sheet.###
>
> Now your task is {task_name}
> ```

