# OpenReview forum: "AgentStore: Scalable Integration of Heterogeneous Agents As Specialized Generalist Computer Assistant"
_ICLR.cc/2025/Conference — ICLR 2025 Conference Withdrawn Submission_

### Official Review · Reviewer_m1su · 2024-10-18

**Soundness:** 3
**Presentation:** 4
**Contribution:** 3
**Rating:** 6
**Confidence:** 4

**Summary:**

AgentStore is a scalable platform designed to integrate diverse agents to automate operating system tasks dynamically. Through its MetaAgent and AgentToken modules, AgentStore achieves state-of-the-art results on the OSWorld benchmark by enhancing adaptability and task execution efficiency.

**Strengths:**

1. I like the scope of this paper. It is necessary to discuss how to scale up by incorporating more evolving agents into one platform.

2. The experiments show the SoTA performances in the OSWorld Benchmark, and the performance is strong compared with other baselines.

3. The figures are interesting. And the claims are straightforward.

**Weaknesses:**

1. I strongly suggest that the authors include at least one additional benchmark. Since the current OSWorld benchmark is relatively new, achieving good results on it may not be fully convincing. Most importantly, it seems that there are many similar benchmarks within the same scope, and incorporating several of them would provide a more comprehensive evaluation.

2. Building on the first weakness, it would be helpful if the authors could conduct experiments that explore the generalizability of the model across different benchmarks.

3. Is the main contribution of this paper training a model that orchestrates different agents, and prior to this, do you also introduce agents in the AgentStore? If so, I believe the most relevant baselines would be models that train to select APIs or tools, which can be analogous to selecting agents (as they function similarly). It would be beneficial to compare your results with these existing methods.

4. Expanding on the 3, I suggest supplementing the experiments using alternative methods to orchestrate the agents within AgentStore. For example, you could compare against RL-based approaches such as [1] GPTSwarm, which orchestrates agents using graphs, or model-based methods like [2] Toolformer, which selects tools from a trained model, and [3] LangChain's ICL-based tool-calling agent.

5. The time or cost analysis of training and inference is missing and would provide valuable insights.

References:

[1] GPTSwarm: Language Agents as Optimizable Graphs." ICML 2024

[2] Toolformer: Language models can teach themselves to use tools." NeurIPS 2024

[3] LangChain: https://python.langchain.com/v0.1/docs/modules/agents/agent_types/tool_calling/

**Questions:**

1. Could you provide more details about the term *hybrid* in Table 1? There are no related explanations in the paper, which makes it unclear for the reviewer to understand the exact meaning of *hybrid* in this context.

2. Is *Hash Manager* a commonly used term in this context? The connection between your paper and *Hash_RC6*, as mentioned, is unclear. Additionally, the statement *"our method narrows the management scope to a few selected agents, leaving ample context space for detailed documentation of these fixed agents. This design shares similarities with hashing methods"* is unclear and could benefit from further clarification.

It would be appreciated if the authors addressed all of the weaknesses and questions.

**Details Of Ethics Concerns:**

No ethics concern for this paper.

---

### Official Review · Reviewer_6VM1 · 2024-10-25

**Soundness:** 2
**Presentation:** 3
**Contribution:** 2
**Rating:** 3
**Confidence:** 3

**Summary:**

The paper introduces AgentStore, a platform designed to integrate and manage a wide variety of digital agents capable of performing specialized tasks on computer systems. This system addresses the limitations of current general-purpose agents, which struggle with complex, open-ended tasks, by using a flexible, scalable approach similar to an app store. AgentStore includes a core MetaAgent that uses a novel AgentToken strategy to dynamically select and manage suitable agents for specific tasks, allowing for collaboration between specialized agents. Experiments show AgentStore's effectiveness on the OSWorld benchmark, significantly outperforming previous systems by more than doubling their success rates on complex tasks. This advancement highlights the potential of AgentStore in developing versatile, specialized assistant systems that improve both user experience and task automation across different environments​.

**Strengths:**

1. AgentStore enables easy integration of various specialized agents, similar to an app store, allowing the platform to continuously expand its capabilities. This adaptability makes it suitable for handling a broad range of tasks in evolving operating system environments.

2. The MetaAgent with AgentToken routes tasks to the most suitable agents and can manage collaborative tasks involving multiple agents. This approach significantly enhances task handling by using minimal resources and avoiding frequent model retraining.

3. AgentStore achieves a marked improvement on challenging benchmarks like OSWorld, doubling the success rates of prior systems. This demonstrates its capability to handle complex tasks across different software and application domains effectively.

**Weaknesses:**

1. The authors claim that their methods "double the performance of previous systems". However, this comparison is not entirely fair, as their approach employs a significantly larger number of agents and incurs substantially higher memory and costs. The paper does not address these additional costs, nor does it include experiments comparing baselines that utilize multiple agents, which would provide a more accurate comparison with the proposed method. I suggest that the authors test multi-agent baselines that use the same group of agents mentioned in the paper.

2. While the authors describe their AgentStore as a "generalist" assistant, the evaluation lacks sufficient breadth. The method could be tested on one additional benchmark such as WebArena or Mind2Web to demonstrate generalizability. Both APPAgent and OSWorld-Multi involve fewer than 100 tasks, which is a relatively small number and could allow for manual tuning of the agents to game the evaluation system.

3. The presentation of the paper lacks rigor. The introduction uses overly fancy language and falls short of the scientific rigor expected, including imprecise terms such as "stunning results." Additionally, in Figure 2, the "AgentPool" is illustrated with agents like Sheet Agent, Slide Agent, Web Agent, etc., which are not clearly defined in the paper. Please provide an explanation of what each of these agents is and how they are built in the main text or appendix, or revise the figure to present a more accurate representation.

4. The related work section is not comprehensive, particularly regarding multi-agent systems. The authors state that previous works "use a fixed number of agents with predefined roles" and that "their agents are usually homogeneous," but this is inaccurate for many studies, such as "Internet of Agents" and "AutoGen". A review of classical papers in multi-agent systems would also reveal that many incorporate heterogeneous agents, a discussion that the authors have entirely overlooked.

**Questions:**

1. I noticed that the number of tasks in the AppAgent paper is higher than those discussed in your paper. Additionally, the accuracy in your paper is reported in increments of "20%," which makes it less convincing, as I didn't see this in the original paper. Did you select a subset of tasks? Please correct me if I'm wrong.

2. Could you make the figures more clear? Currently, there are too many elements, especially in Figure 2, making the figures look cluttered.

3. For AgentMatch, you mention a "ground truth set of agents required for successful task completion." What if multiple different sets could successfully complete the tasks, making it so there's no single ground truth?

---

### Official Review · Reviewer_JEbG · 2024-10-29

**Soundness:** 2
**Presentation:** 2
**Contribution:** 2
**Rating:** 3
**Confidence:** 4

**Summary:**

This paper proposes a method for managing and deploying multiple different agents to achieve computer control tasks. The approach collects a group of agents, AgentStore, each with different capabilities and domain specialties. Each agent has an associated document describing the agent.

In order to deploy the correct agent for a given task, the paper uses “AgentToken”, which is a trained embedding for selecting an appropriate agent to deploy for the task. For more complex tasks, the AgentManager can select up to k tasks. The paper demonstrates SoTA performance on OSWorld. They also release a dataset, based on OSWorld for tasks that require multiple agents.

**Strengths:**

The paper proposes an interesting idea of combining together multiple agents to solve complex tasks. There is a clearly significant engineering effort that went into creating this work, i.e. to create nearly twenty different agents and documents from scratch. The approach achieves impressive performance on a very challenging benchmark.

**Weaknesses:**

The paper may overstate the difficulty of agent selection and potentially understates how much the success depends on designing customized agents for the applications specifically in the benchmark. While MetaAgent with AgentToken is presented as a main contribution, the paper does not conclusively demonstrate its superiority over an ICL baseline.

The reported 49.63% accuracy for ICL with GPT-4o in agent routing seems unusually low. This appears to stem from implementing a simplistic ICL baseline that inefficiently includes entire agent documents in the prompt (as shown in the appendix, though the baseline implementation should be better detailed in the main text). A fair comparison would require:
* Testing ICL with more concise capability descriptions
* Including few-shot examples
* Providing concrete examples demonstrating where ICL fails compared to AgentToken

The system's current implementation raises significant scalability concerns:

* Custom agents and documentation were developed specifically for each app in the OSWorld dataset
* Scaling requires substantial manual effort for each new application:
  * Collecting demonstrations
  * Implementing new agents
  * Writing documentation
* The strong performance appears largely attributable to carefully engineered custom agents rather than a scalable automated approach
* True scalability would require automated agent generation

The paper lacks sufficient detail on the nearly twenty custom different agents (excluding existing ones e.g., Friday) used in the system. Without these details, it is difficult to assess the effectiveness of the approach. Most concerning is the unclear origin of training demonstrations and their potential overlap with OSWorld test tasks. The paper should:
* Specify the source of demonstrations
* Detail measures taken to prevent data leakage between training and test sets
* Discuss how generalization is ensured

There are spelling and grammar errors.
* In Figure 1,”SildeAgent specialize…” should be “SlideAgent specializes…”,
* In Figure 1, “are required to collaborate system-wide” should be “are required to collaborate on system-wide…”
* In the prompts in the Appendix, “Demostation” should be “Demonstration”
* In the prompts in the Appendix, “Templete” should be “Template”

**Questions:**

Please see weakness for points of clarification desired.

---

### Official Review · Reviewer_tQJj · 2024-11-02

**Soundness:** 1
**Presentation:** 2
**Contribution:** 2
**Rating:** 5
**Confidence:** 4

**Summary:**

This paper presents AgentStore, which allows integrating and dynamically use a range of domain-specific agents (the difference is mainly the base model, how the model is prompted, the action/observation space for each agents).
They train a model, named MetaAgent, to dynamically select the agents and distribute the tasks given current context.
Performance is verified OsWorld.

**Strengths:**

1. The idea of effectively integrating and dynamically use a range of domain-specific agents is interesting.

**Weaknesses:**

1. Vague Agent Descriptions: The descriptions of agents in the AgentPool are too insufficient to understand what each agent actually are. The only available information seems to be from Table 6, which provides only names for many agents. What distinguishes each agent, such as SheetAgent from SlideAgent? Is it simply their prompts, or are there other differences?
2. Over-engineered, Overfitting to OSWorld: Many agents in Table 6 appear optimized for tasks specific to OSWorld, raising doubts about their general applicability. Evaluating the system against broader benchmarks, like GAIA or SWE-Bench, would strengthen the claim of generalist capabilities.
3. Scalability Concerns: the claimed scalability of this system is unclear. Will there be contributors create specialized agents? And can this platform effectively integrate diverse agents? Table 6 shows that 18 of the 20 agents are authored by the team. So it's unclear if this system design can effectively integrate diverse agents found in the wild.
3. Missing Key Baselines in AgentToken: The study only presents AgentToken training with tunable embedding layers. It would be valuable to compare performance and efficiency when the entire model is tunable to understand the trade-offs better.

**Questions:**

1. How are AgentStore(FT) baseline constructed? Why is it performing worse then AT?

---

### Note · Authors · 2024-11-22

**Comment:**

Thanks to all the reviewers for their feedback. Due to time constraints, additional results and analyses require more time to complete. We will continue to improve our work！

**Withdrawal Confirmation:**

I have read and agree with the venue's withdrawal policy on behalf of myself and my co-authors.